# The Use of *Gonimbrasia belina* (Westwood, 1849) and *Cirina forda* (Westwood, 1849) Caterpillars (Lepidoptera: Sarturniidae) as Food Sources and Income Generators in Africa

**DOI:** 10.3390/foods12112184

**Published:** 2023-05-29

**Authors:** Lufuno Ethel Nemadodzi, Gudani Millicent Managa, Gerhard Prinsloo

**Affiliations:** Department of Agriculture and Animal Health, University of South Africa, Private Bag X6, Florida 1710, South Africa

**Keywords:** mopane worm, *Cirina forda*, food security, African countries, nutrition, income generation

## Abstract

*Gonimbrasia belina* (mopane worm) and *Cirina forda* caterpillars (Lepidoptera: Saturniidae) are mostly found in shrubs and trees, from where they are collected as larvae and are widely consumed across southern Africa by rural and increasingly urban populations. These caterpillars are among the most prominent, traded, and economically beneficial edible insects found in Western African countries, but also in South Africa, Zimbabwe, Botswana, and the Democratic Republic of the Congo. Over the years, these caterpillars have evolved from being part of the diet in various communities to playing a vital role in income generation. In addition, consumption of *G. belina* and *C. forda* caterpillars as potential food sources has gained momentum due to their potential for contributing to livelihoods and mitigating food security challenges across Africa while providing significant benefits to developing countries on a socio-economic and ecological level. Edible caterpillars serve as a good source of rich nutrients such as proteins, fatty acids, and micronutrients and can be used in formulating nutrient-dense complementary foods. However, limited information is available, specifically on different trees that serve as hosts to these caterpillars, as they depend on the leaves as their only source of food. In addition, the review aims to critique and document knowledge on the nutritional benefits, acceptance of the use of these caterpillars as food security, commercial value, and level of acceptance towards the utilization of caterpillars as food sources.

## 1. Introduction

Edible insects are of great importance in the development of poor populations that exploit them [1] and have been consumed for centuries in African countries, generally by black Africans. However, there is a wrong perception that links eating insects with poverty rather than as a means to sustainability, food security, and income generation.

The collection of edible insects from the wild for food is not a new phenomenon. It is a very old practice that was performed before the development of agriculture, when hunting wild animals and harvesting plants were relied upon by humans [2]. A study Nonaka [3,4] reported that the practice of collecting edible insects in the wild has played a significant role in shaping the history of human nutrition for many cultures over centuries as well as providing ecological and socio-economic benefits for developing countries. Another study Badanaro et al. [1,5] reported that globally, approximately 2000 insect species are edible and are consumed by more than 2 billion people; within the insect orders, Coleoptera (beetles), Hymenoptera (wasps, ants, and bees), Orthoptera (grasshoppers, crickets, and locusts), and Lepidoptera (caterpillars), which constitute 80% of edible species. However, a recent study [6] questioned the above-mentioned 2 billion figure on the number of people who consume edible insects and stated that it was overexaggerated, indicating the difficulty in determining the exact number of edible insect consumers. In Africa, about 500 species of edible insects are consumed, and they form part of the traditional diet. Of these 500 species reported to be consumed in Africa, 256 species are consumed in the Central African region, 165 in southern Africa, 100 in eastern Africa, and 91 in western Africa, while only eight species are consumed in northern Africa [7]. Mabossy-Mobouna et al. [8] identified 24 from the Ngotto forest in the Central African Republic (CAR), Latham [9] identified 28 edible caterpillar species across the Republic of Congo, while Latham (2015) [10] documented 35 edible caterpillars in the Bas-Congo province of the Democratic Republic of the Congo (DRC). Furthermore, Balinga et al. [7,11] indicated that in Africa, approximately 60% of the edible insect species belong to the Lepidoptera, Saturniidae, Sphingidae, and Notodondontidae, which are the three major families of Lepidoptera. In the Haut-Katanga province of the Democratic Republic of the Congo, over 90% of the population depends upon natural resources for food, with more than 65 species of edible insects reported as food in at least 22 families, whereas only 28% of the inhabitants consume edible insects, especially caterpillars [12,13].

Saturniid caterpillars are reported to widely occur across Africa, Australia, Asia, and the United States of America, while their pattern of consumption and availability has been attested in the West [1,14,15]. Saturniidae larvae are popular among Africans because they are large and plentiful. When they first appear, they are hand-picked and collected in large quantities to be exported to the sub-region and to diasporas living in Europe [16,17]. *Cirina forda* and *G. belina* (Westwood, 1849), both belonging to the order Lepidoptera, are the most popular and recognized edible species of the Saturniidae family in Africa, particularly in Nigeria, Zambia, Zimbabwe, Botswana, South Africa, the Central African Republic, and the Democratic Republic of the Congo. A study [11,18,19,20] highlighted that Saturniid caterpillars can be identified by large larval forms with spines on the surface that pupate into cocoons on plants or leaf litter in the ground, from which vividly colored moths emerge [21]. Moreover, these edible Saturniid species are consumed by various African tribes for their taste, cultural significance, and nutritional value, as well as supplementary food when staple foods are scarce [22].

The larval stage of the emperor moth *G. belina*, often known as “*phane*” in Botswana, is the mopane worm, which belongs to the Saturniidae family [23], which derived its name from *Colophospermum mopane* trees. The *Colosphospermum mopane* (Benth, 1865) tree is the major host for *G. belina*, whose outbreaks are reported to totally defoliate large tracts of woodland in six weeks during the early growing season and occasionally for a second time later in the growing season [24,25]. The caterpillars are also known by other vernacular names; for instance, in South Africa, they are referred to as *Mashonzha* in TshiVenda and *Masonja* in Sepedi; in Zimbabwe, the Shona tribe calls them *Madora*, the Kalanga tribe calls them *Mahonja,* and the Ndebele tribe calls them *Amancimbi*. The mopane worm larvae can be identified by their black spines, which are red, grey, or green in color. The reddish-brown *G. belina* moth is characterized by a brown eyespot on the hindwing, which is surrounded by black and white rings; the forewing has a small glass speck. The hindwing’s anterior portion is reddish in color. Each year, normally in the months of December, March, and April (Rasiwelame, O., 13 February 2022), the larvae are hand-picked in large quantities in different regions.

*Cirina forda* is commonly known as the pallid emperor moth or shea defoliator [26]. The pallid emperor moth, or shea defoliator, is another edible Saturniid larva that is well accepted as food with high commercial value in Africa [1]. *Cirina forda* caterpillars are collected, processed, and consumed for other reasons, such as medicinal properties, aside from their nutritional benefit. It is reported that the pallid emperor moth larvae are extensively consumed and are in great demand as it has become the most commercialized species in Nigeria, while in Togo, the Savannah region is involved in the consumption and sale of *C. forda,* locally known as “*Salantonda*” by the Moba ethnic group [27,28,29]. In South Africa, the *Burkea africana* (Joseph Burke, 1843) tree is famously known to host *C*. *forda* caterpillars and is the most preferred source of food for these caterpillars [30]. *Burkea africana* is predominant in the Limpopo, Gauteng, and Mpumalanga provinces of South Africa. *Cirina forda* larvae have two color forms, ranging from black with yellow bands to black with white bands. The hindwing has a little black eyespot and is light brown in hue [10].

In South Africa, *G*. *belina* and *C*. *forda* are the most consumed caterpillars, specifically by the Vhavenda, Bapedi, and Vatsonga tribes, which reside in the Limpopo province, the northern region of South Africa [31]. Recently, an increase in relocation and/or interprovincial relocation to urban areas and cities has created awareness of the edible caterpillars among other tribes, such as the Amandebele of Mpumalanga province.

Several studies have been conducted to investigate various elements of using insects for food [32,33,34]. However, only a few studies have focused on indigenous knowledge, educational levels, gender participation, ethnicity, preferences, acceptability, and consumption of edible insects as food to enhance food security. Therefore, the review aimed at summarizing information on the use of mopane worms and *C. forda* caterpillars to aid in understanding the role of edible caterpillars in their nutritional, economic, and food security benefits. Additionally, the review documents the relationship between host trees and edible caterpillars, i.e., mopane worm and *C. forda,* that feed on the leaves of these trees, as well as the different locations of these host plants.

## 2. Materials and Methods

Various databases were consulted in searching for suitable literature, such as Science Direct, accessed via the University of South Africa online library, Google, Research Gate, and Google Scholar. String searches were conducted using the search terms “edible insects, edible caterpillars, mopane worms, *Gonimbrasia belina*, *Cirina forda*, edible security, food security, and edible insect sustainability”. Inclusion criteria required that information be scientific, peer-reviewed publications, including books, dissertations, and theses dealing with edible insects, edible caterpillars, and specifically the two specific caterpillar species. Exclusion criteria were minimal since limited publications were found using the search words and all peer-reviewed sources were consulted with no limitation on years of publication. Popular publications and web-based information that was not considered peer-reviewed were excluded to ensure the credibility and trustworthiness of the information reported.

## 3. Results and Discussion

### 3.1. Occurrence and Collection of Edible Caterpillars

Saturniid species have proven to inhabit a variety of bio-ecological zones with significant seasonal variation and have a high level of specialization and preference for diverse host plants [35]. As with other immature stages of Lepidoptera, caterpillars rely on the leaves of plants, especially trees, for nourishment; consequently, their availability depends on the occurrence and livelihood of these host plants [11]. Muvatsi et al. [36] reported that in southern and eastern Africa, caterpillars are known to occur in huge epidemics in arid and savannah environments, depending on the region.

Upon infestation, caterpillars feed on the leaves of the host plant until physiological maturity is reached, after which they fall onto the ground as nymphs, ultimately dying and decomposing into the soil [31]. Other findings suggest that the mature caterpillars move across the land in search of a good area to bury themselves in the soil [37]. A recent study conducted has revealed that, during the beginning of the rainy season, many caterpillars cause severe defoliation on their host trees while consuming young leaves [31]. Since the trees develop new leaves following defoliation, these cause relatively minor damage to the host plants [8,37].

The collection of edible insects from the wild for food is a very ancient practice and has played a significant role in shaping the history of human nutrition for centuries in many cultures [2]. In South Africa, *G*. *belina* and *C. forda* occur twice a year, from December to January and again in March/April to May, and are harvested when they reach the 4th and 5th instar stages of maturity [38]. The period of collection depends on the area, when a new flush occurs, and ultimately the time at which the moth lays eggs (Figure 1a,b). For instance, in Mpumalanga and Gauteng provinces in South Africa and Botswana, a new flush occurs in August–September, egg laying happens in October, the hatching of the eggs takes place in late October, and small, tiny larvae become visible in November (Figure 1c), and mature caterpillars are only visible in November, followed by the collection of these caterpillars as larvae in December (Figure 1d) (Raletsana P., 13 February 2022, personal communication). On the contrary, in Limpopo province of South Africa, new flushes only happen in January–February; egg laying occurs in February, followed by hatching in March, with small, tiny larvae visible in March, according to Rasiwelame O. (13 February 2022). Nymphs (mature caterpillars) are only collected in April. A telephone interview with an edible caterpillar collector from Botswana who has made a career out of collecting edible caterpillars as a supplier and distributor revealed that after the eggs have hatched, it takes 3 weeks for the larvae to develop into fully mature caterpillars. Collection starts from 6 a.m. until 6 p.m. and is performed manually. On average, collectors fill a 50-kg sack with fresh *C. forda* and *G*. *belina* caterpillars per day per person. He also mentioned that *G*. *belina* and *C*. *forda* highly rely on the rain to fully develop. Absence of rain or drought usually means that the larvae would not develop as the trees would not have flush leaves, which is their only source of food.

In the study conducted in Nigeria by [28], it was highlighted that the shea butter tree, *Vitellaria paradoxa*, is the solitary host for the *C. forda* larva in Nigeria as well as other West African subregions. Moreover, during the months of May and June, eggs are found on the host plants, and the larvae are harvested between June and August, while once a year, in the months of July and August, the larvae are harvested in large quantities from the shea butter trees [39]. In Kenya, it was discovered that *C. forda* has a bivoltine life cycle, with larvae appearing in April–June and October–December. It was also observed that the larvae of *C. forda* in the DRC are bilvotine, where they are available between July and September in Western Kasa; September–December in Ban-dundu; and June–September in the region of Kisangani, with different caterpillar species appearing on their host trees in succession [40] and each species only present during a given period of the season. *Cirina forda*, on the other hand, has been shown to be univoltine in Togo and Nigeria, with nymphs occurring between July and September [1,41]. Mature females lay eggs in clusters on the twigs of host trees, and larvae emerge in July after a 30- to 35-day incubation period. The larvae aggregate and feed for about 6 weeks before pupating in the soil for 9 months [41]. The ability of larvae to locate food soon after emergence is critical to their survival. According to a study conducted in the Democratic Republic of the Congo, the larvae start appearing between November and January [10]. These larvae are gathered from the leaves and collected in pitfall traps set around the base of the trees, where the descending larvae are trapped [28]. Larval aggregation and feeding damage the host tree’s leaves significantly [31], as shown in Figure 1d.

### 3.2. Host Plants/Trees of Gonimbrasia Belina and Cirina Forda Caterpillars

As an alternative, certain host tree species might be promoted in forest reforestation or agroforestry programs to benefit local communities from both caterpillars and timber collection. Local residents in rural areas of the tropics rely on forests for a variety of food resources, such as firewood and medicinal purposes, which are also among the most vital timber species collected. On the contrary, caterpillars rely on these trees as their hosts and only source of food.

The mopane worm larvae have been recorded feeding especially on *Colophospermum mopane* Kirk ex J. *C*. Léonard tree, which belongs to the family Fabaceae [42], while *Vittelaria paradoxa* C.F. Gaertn of the Sapotaceae family is highly preferred by *C. forda* in West African countries [43,44]. In a survey performed by [11], it was highlighted that across the many sites surveyed, the caterpillars were discovered to have unique host plant specificity. It was observed that mopane worms mainly feed on *Anacardium occidentale* Lin (Anacardiaceae) and *Mangifera indica* L. (Anacardiaceae).

*Cirina forda* appeared to have a greater host range than *Euclea divinorum*, *Manilkara sulcata* Engl, and *Acacia mearrnsii* De Wild (Fabaceae) [11,45]. Surprisingly, in a recent study, it was noted that *C. forda larvae* were found feeding on *Manilkara sulcate*, *A. mearnsii,* and *E. divinorum* in Kenya, while it is known to feed mostly on *Crossopteryx febrifuga* [10] in the Democratic Republic of the Congo (DRC).

On the contrary, in West Africa, C. *forda* is only found feeding on *V. paradoxa* [10], while in southern Africa, the host trees comprise *B. africana* [32] Hook and *Albizia versicolor* Welw. ex Oliv., both of which belong to the Fabaceae family [22]. Ande, Fosarati [40] reported that it is notable that larvae of *C. forda* feed on the leaves of the tree species Sapelli (*Entandrophragma cyclindricum*) and Tali (*Erythrophleum suaveolens*), which are harvested for timber in the DRC. In the Congo Basin, Tali ranks fourth in volume among them. In Cameroon, caterpillars are reported to be a very important source of food. A survey conducted by [17] in the Centre region of Cameroon (Yaoundé) highlighted that *C. forda,* which is locally known as *Minlone*, feeds on the leaves of *Erythrophleum suaveolens* and *Baillonella toxisperma* tree species, and the larvae are only collected from June to September*. Burkea qfricana,* commonly called *Mufhulu* in Venda and *Mpulu* in Tsonga, is a host plant for *C. forda* caterpillars, which feed and cause severe defoliation on the leaves in South Africa [46].

### 3.3. Income Generation

In developing countries such as South Africa [16], rural communities greatly rely on indigenous natural resources. For instance, edible insects and wild fruits [47,48,49] are primarily used for household consumption and to generate income through trading to sustain livelihoods. Furthermore, the study conducted in central Cameroon highlighted that there are over 300 species of edible caterpillars in the world, of which over 130 are consumed in Africa and some are commercialized, providing significant income generation to local communities [17]. Defoliart [50] reported that approximately 1600 tons of caterpillars (*G. belina*) were sold annually in South Africa, as shown in Table 1.

Mopane worms have been commercialized and contribute significantly to rural economies, with 63% of collected and processed larvae in Limpopo, South Africa, sold in local marketplaces [38]. For instance, in Vhembe district, local informal marketplaces in Thohoyandou, Sibasa, and Tshakhuma are famously known to be the economic hub for trading *G*. *belina* and *C*. *forda,* as shown in Figure 2a,b. In the trade industry, unemployed women are usually at the forefront of the collection and sale of these caterpillars, and the income generated is used to sustain their families and pay for their children’s school and university fees. In South Africa, trading of *G*. *belina* and *C*. *forda* is usually performed in different phases; for instance, the collectors, also referred to as suppliers, sell in large quantities to the distributors, who sell to the consumers in different small packages depending on the quantity needed and affordability.

However, in the year 2022, the prices differ from 80 kg, which is sold for R6000.00, to 50 kg, which is sold for R3000.00, and 25 kg, which is sold for R1800 (Masethe V., 13 February 2022, personal interaction). The distributors, who sell to the small emerging vendors, sell 20 L for R1000, and 5 L are sold for R300. The small, emerging vendors sell to the consumer in small packets, equivalent to a small beaker size, for R30. This has brought in different ways of income generation, which has led to the alleviation of poverty in rural communities. Recently, all the Spar grocery stores in Vhembe District are selling *C*. *forda* and *G*. *belina* at prices ranging from R19.04–R24.99 per small packet per 0.112 kg and R169.99 per kg, as shown in Figure 3, which is an experiment for commercialization in South Africa for trading these edible caterpillars in top food retail markets.

Several researchers have reported that, because caterpillars have high nutritional content, they are in high demand, especially in the rural areas of southern Africa; additionally, they are highly sought-after delicacies in other countries, including Malawi and Zambia, and through trade, they have also contributed to people’s livelihoods [40,56]. In the early 1980s, tons of caterpillars were noted to be collected per year, with [57] estimating that around 1,600,000 kg of mopane caterpillars were sold, as per the South African Bureau of Standards (SABS). The dry mass of mopane caterpillars grew in value from less than US $0.5 (R8.51)/kg in the early 1980s to more than US $10 (R170.16)/kg by 1996 [58]. Many areas in South Africa place high value on mopane worms, which play a role in an unsustainable manner of collection and non-selective harvesting that has resulted in the extinction of numerous *G. belina* populations [59,60]. The high demand for mopane worms in South Africa has resulted in overexploitation, importation, and outsourcing from neighboring countries such as Zimbabwe [26,59] and Botswana [61].

### 3.4. Food Security

In most African countries, food security remains one of the most fundamental issues for economic growth and human health due to insufficient food availability and accessibility to meet the nutritional needs of their people [33]. In recent years, many cases of insufficient food have been recorded. Oberprieler [20] reported that food security issues arise due to massive expansion in the global human population, which is expected to reach approximately 9 billion people by 2050, resulting in a 70% increase in food demand and an increase in food prices, while the rise in food prices will push people to look for low-cost, long-term protein sources that can be found in the forest. Bayeigunhi, Oppong [35] highlighted that in sub-Saharan Africa, most of the population resides in rural regions, where the main source of livelihood is agriculture, while equally rural and underprivileged urban areas are becoming increasingly reliant on the forest for a living due to high rates of poverty and a lack of food. Over the centuries, throughout the world, non-timber forest products have been key aspects of household diets, with many people dependent upon a wide range of forest products, classified as timber and non-timber forest products (NTFPs), to maintain their livelihoods [49]. Within rural communities, where poor households collect, process, and exchange natural resources from forests, woodlands, grasslands, and rivers, they are more reliant on the forest to obtain both food and income from NTFPs, as it is reported that these products also provide proteins, minerals, and vitamins, as well as adding diversity to local diets [40,62].

In Central Africa, forests supply food and other subsistence products to approximately 69 million people who live within and around these forests. In addition, the forest contributes to the nutrition of an additional 40 million people who dwell in urban areas near the forest estate [40]. Among the important foods obtained from forests are edible caterpillars and/or insects, which are both consumed and sold, contributing to the food security and income of the rural and urban populations [35]. Edible insects have been suggested as one of the solutions to improving food security and reducing poverty by the Food and Agriculture Organization of the United Nations [7]. In addition, recent studies have shown that edible insects are rich sources of antioxidants and have also been described as an alternative protein source since they are an important protein-rich natural resource [11,63], but they also play a vital traditional role in nutritious diets in various countries in Africa [7,64]. In recent years, the consumption of edible insects has received more attention due to their potential to alleviate food security problems around the world and their contribution to livelihoods [65,66]. Furthermore, human consumption of edible insects is proven to be an environmentally benign approach to increasing food consumption, and it could be a viable solution to food shortages. In Africa, China, Thailand, Japan, Latin America, and Mexico, edible insects are consumed as food, and about 2 billion people globally frequently consume edible insects as part of their diets [20].

Mopane worm, *G. belina,* is the most economically important *Saturniid* that is widely consumed in southern Africa. The larvae of *G. belina* are a valuable food source for rural inhabitants living around the mopane woodland, and they even served as delicacies, especially in the *Mopane* belt, which stretches over Angola, Zambia, Zimbabwe, Mozambique, South Africa (Figure 4), and Botswana [11,55]. According to [67], Africa has the lowest protein intake per capita per day, and given the continent’s severe financial constraints, particularly in rural areas, the availability of alternative protein sources is critical. These caterpillars are significant to rural people’s livelihoods because they have three times the protein content of beef per unit weight and can be stored for many months once dried [40]. Furthermore, they are a valuable trading commodity that is an instinctive part of local cultural practices.

Edible *Saturniid* caterpillars are highly nutritious, and their nutritional values have long been recognized [68]. In Africa, edible insects are frequently consumed and serve a vital role in a healthy diet; however, insect choice and consumption vary by species and order [69]. Furthermore, in rural parts of southern Africa, Uganda, and Nigeria, various caterpillar species play a significant role in generating income [19]. Caterpillars of the order Lepidoptera are the most popular, and they are the most chosen species due to their nutritional value, which includes protein, lipids, and important micronutrients [69,70] (Table 2). Processed mopane worms are reported to contain about 60% crude protein, 17% crude fat, and 11% minerals such as calcium, iron, and phosphorus [58,71]. Since most caterpillars have a significant quantity of iron, they are frequently given to pregnant and nursing mothers, as well as people suffering from anemia, to help increase the amount of iron, calcium, and protein in their diet [33,40]. These caterpillars have also been reported to be very rich in key vitamins such as vitamin B1, vitamin B6, vitamin B12, thiamine, and riboflavin [37,72,73]. In addition, high unsaturated fatty acids (Table 2, particularly linoleic and linolenic acids, were reported to be higher in *Saturniid* caterpillars than all animal oil sources and palm oil, which is a plant derivative [74]. It was highlighted that since the activity of the 6Δ desaturase enzyme decreases with age, it may be beneficial in the diet of the elderly. Apart from its dietary value, the oil can be used in pharmaceuticals and other sectors [15]. Subsequently, edible caterpillars help to prevent malnutrition and are a significant component of young children’s diets, as they are necessary for their physical and cognitive development [39].

A study Henley et al. [75] reported that *C. forda* has a 14% fat content and a protein content (Table 3) of more than 20%, which is comparable to beef at 26%. It is therefore a reliable protein food source, and a low level of cholesterol has also been observed. A nutrient composition of 54 edible insects based on dry matter was compiled in a study conducted by [19], focusing on the Lepidoptera order. The findings highlighted that the Lepidoptera had the highest protein (12–79%), crude fiber (2–16%), and fat (2–55%), while moisture content was 3–86%. Moreover, Lepidoptera showed the highest micronutrient content, such as phosphorus (100–730 mg/100 g) and magnesium (1–160 mg/100 g). A study by [19] found that edible insects have low levels of vitamins A, B2, and C. The 100 g dry matter of edible insects highlighted in the study did not provide enough of the daily required vitamin A of 500–600 mg or vitamin C of 500–600 mg. However, Alamu et al. [4] reported that the excellent source of vitamins found in some edible insects has a lot of potential to be utilized as a healthy food supplement for malnourished people or to prevent malnutrition (Table 4).

Edible insects have a high protein content, which could assist in the battle against protein deficiency, a major contributor to human malnutrition [76], which is the most common cause of malnutrition in Africa and must be managed to prevent starvation [77]. Since caterpillars are known for their numerous nutritional benefits, some African countries have practices that include the grounding of dried caterpillars into flour to fight malnutrition in children. However, caterpillar milling is reported to negatively affect the nutritive value. It is evident from Table 2 that processed edible caterpillars are rich in sodium in comparison to the fresh ones. On the contrary, minerals such as potassium (K), iron (Fe), and magnesium (Mg) are highly abundant in edible caterpillars. However, differences in calcium and phosphorus between processed caterpillars and raw caterpillars were found not to be statistically significant. Processed or not, *C*. *forda* caterpillars are a good source of protein and minerals [1].

In situations where there is a scarcity of high-quality foods, the inclusion of *G*. *belina* and *C*. *forda* caterpillars in regular diets may help reduce the rate of malnutrition [34,63]. Therefore, as a result, entomophagy is increasingly gaining popularity, not only in Africa, Asia, and Latin America, where this practice is reputable [13,78,79], but also in other parts of the world such as Europe [80,81,82,83].

**Table 2 foods-12-02184-t002:** Moisture, ash, crude protein, crude fiber (%), and mineral (mg/100 g) contents of fresh and converted *C*. *forda* [1].

Analysed Parameters	Raw Caterpillars	Processed Caterpillars
**Proximate composition (%)**		
Moisture	10.06	3.94
Proteins	51.43	52.39
Fibre	8.42	8.92
Ashes	18.43	16.48
**Minerals (mg/100 g)**		
Sodium	35.035	39.838
Potassium	67.386	43.124
Calcium	28.531	28.356
Magnesium	24.472	23.638
Iron	6.929	5.061
Zinc	1.068	0.734
Manganese	1.751	0.972
Copper	0.454	0.341
Phosphorus	233.09	232.87

According to Table 3 below, the proximate composition of *C. forda* larvae revealed a protein content of 33.12%, which is within the protein range of 15–60%, and 7.12 g/100 g of ash content, which agrees with the findings reported by [79]. Furthermore, the findings indicate that both *C*. *forda* and *G*. *belina* are great sources of fat, ranging from 12.24% to 16.37%, respectively. The composition of *G. belina* is highly comparable to that of *C. forda* for protein, fat, ash, and crude fiber.

**Table 3 foods-12-02184-t003:** Chemical composition of *Cirina forda* and *Gonimbrasia belina* larvae [84,85].

Element	*C. forda*	*G. belina*
Protein	33.12 %	55%
Fat	12.24%	16.37%
Ash	7.12%	5.8%
Crude Fibre	9.40%	27.8%
Carbohydrates (by difference) Total Polyunsaturated fatty acid Total Monounsaturated fatty acid Total Saturated fatty acid Energy	38.12% 53.80 14.60 31.60 359.60 (Kcal/100 g)	13.8%

**Table 4 foods-12-02184-t004:** Comparison of mineral composition in *C. forda* (mg/g dry matter) and *G. belina* (mg/g) [84,85].

Element	*C. forda* (mg/g)	*G. belina* (mg/g)
Calcium	0.07	16.0
Potassium	21.30	35.2
Magnesium	0.324	4.1
Phosphorus	10.90	14.7
Sodium	2.10	33.3
Iron	0.64	12.7
Zinc	0.086	1.9
Manganese	0.07	

The larvae of *G. belina* and *C. forda* are rich in mineral elements, although higher contents are reported in *C. forda* (see Table 4). For instance, 100 g of dry sample is reported to contain 32.4 mg Mg, 8.6 mg Zn, 64.0 mg Fe, and 1090 mg P. According to Mureki et al. [86], consumption of 100 g of dry larvae is reported to provide 100% (P), 355% (Fe), and 57% (Zn) of the Recommended Dietary Allowances (RDA). In addition, Na and K in the larva are reported to contain 210 mg/100 g and 2130 mg/100 g levels, respectively, which result in a K-to-Na ratio of approximately 10:1. Including *C. forda* larvae as a potential component of diets with a 1.7 K/Na ratio is inferred to aid in the management of hypertension and certain coronary heart diseases [83,87]. Furthermore, the intake of K is reported to reduce blood pressure by alienating the biological effect of sodium [88,89].

Due to their excellent nutritional composition, edible caterpillars are currently serving as food resources and/or protein sources for humans on a daily basis (see Figure 4). Palmitic (13.0%), myristic (0.7%), and stearic (16.0%) acids are the major fatty acids present, while 31.6% of fat is made of saturated fatty acids, which is comparable to 35.5% and 29.6% reported for poultry and fish but lower than 52.0% and 44.1% reported for beef and pork, respectively [54]. The above-mentioned saturated fatty acids have been demonstrated to raise low-density lipoprotein (LDL) cholesterol and are considered atherogenic [90]. However, a study by [83] indicated that nearly 50% of the stearic acid in the larva has been reported not to raise plasma LDL cholesterol. Of the 68.6% of unsaturated fatty acids, only 14.6% and 95% of the oleic acid account for monounsaturated fatty acids, which are hypocholesterolemic [91]. Furthermore, larvae contain a high amount of polyunsaturated fatty acids (8.1% linoleic acid and 45.3% linolenic acid). The polyunsaturated to saturated fatty acid (P/S) ratio has been extensively used to indicate the potential to lower cholesterol in food. For instance, a 0.2 P/S ratio has been linked to high levels of cholesterol and a high risk of coronary artery disease. On the contrary, a high ratio of 0.8 is reported to be associated with a required cholesterol level and low coronary heart disease [92].

## 4. Conclusions

The current review documented seasonality, the different trees that serve as hosts for *G. belina* and *C. forda* caterpillars, their economic value in rural communities, and their role in food security. Indigenous knowledge is pivotal and should encourage communities to preserve indigenous trees while protecting habitats for edible *Saturniid* species. In addition, this review’s findings should be used as a good source of collective findings on local plants, which are vital when collecting caterpillars. This manuscript reports that *G. belina* and *C. forda* have the potential to provide significant amounts of minerals, polyunsaturated fatty acids, and proteins to low-income people whose diets are generally inadequate in animal protein. In addition, a high potassium and sodium ratio of 10:1 and 1.7 polyunsaturated fatty acids to saturated fatty acids is reported in larvae. Furthermore, this review provides an insight that will help comprehend the inter-annual fluctuations that may lead to the decrease in abundance of caterpillar species’, reported to be attributed to logging of host trees, climate change, and the cutting of tree organs for divine reasons or medicinal purposes. It is therefore advisable to encourage communities to protect different tree species that serve as hosts for edible caterpillar species, particularly during seasonal outbreaks, which will ultimately ensure food conservation and food security with a high nutritional composition across African countries and the world at large. To assure a consistent supply of edible caterpillars, more research on the prospects for mass production through inoculating growth-promoting metabolites to influence effective growth of *B*. *africana* trees outside their natural habitat, as reported by [29], and *C. mopane* trees as hosts is required to avoid over-collection, which will prevent overexploitation of these insects in the wild forest in order to improve constant supply and sustainability to use insect biodiversity and promote ethno-entomophagy.

## 5. Recommendation

Although several studies on *C*. *forda* and *G*. *belina* have been conducted for their nutritional benefits, a few studies and findings still need to be confirmed on their medicinal properties and/or health benefits for humans. Further research on the cytotoxicity of raw *C. forda* and *G. belina* must also be carried out to profile the risk associated with the consumption of caterpillars. Since the collection of caterpillars is not performed in a hygienic manner, there is a huge possibility of contamination of the caterpillars by soil microorganisms, which warrants future research on identifying fungi and/or bacteria that may be harmful if consumed in large quantities. Lastly, more studies on the symbiotic relationship between *B*. *africana* trees and *C*. *forda* ought to be carried out to further emphasize the synergies and the role *C*. *forda* plays in the growth of *B*. *africana* trees. To date, there is a huge gap in the commercialization of edible caterpillars that needs to be explored to ensure continuous availability for consumption.

## Figures and Tables

**Figure 1 foods-12-02184-f001:**
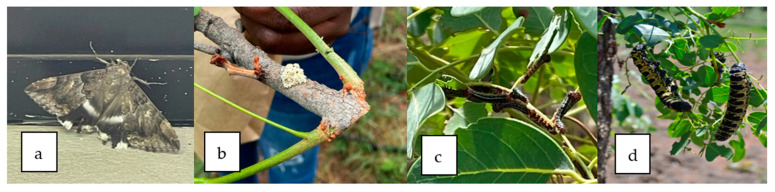
Stages of the life cycle of *Cirina forda* showing (**a**) castor semi looper moth, (**b**) egg clusters, (**c**) instar I, and (**d**) instar V, or mature caterpillar (Nemadodzi L.E., October 2021).

**Figure 2 foods-12-02184-f002:**
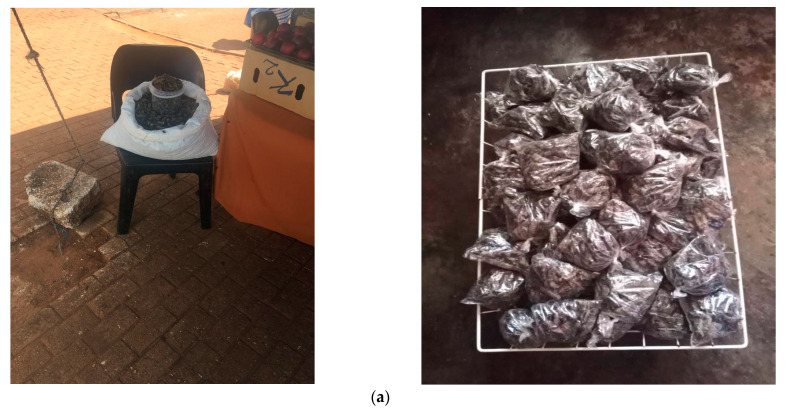
(**a**): Standardized weights of dried caterpillars sold in various-sized containers and small plastic bags, used to calculate the unit-selling price by a small emerging edible caterpillar seller/vendor in Thohoyandou (Nemadodzi L.E., January 2022). (**b**): Standardized weights of dried caterpillars sold in plastic mugs were used to calculate the unit selling price by a small emerging edible caterpillar seller/vendor in Thohoyandou (Nemadodzi L.E., January 2023).

**Figure 3 foods-12-02184-f003:**
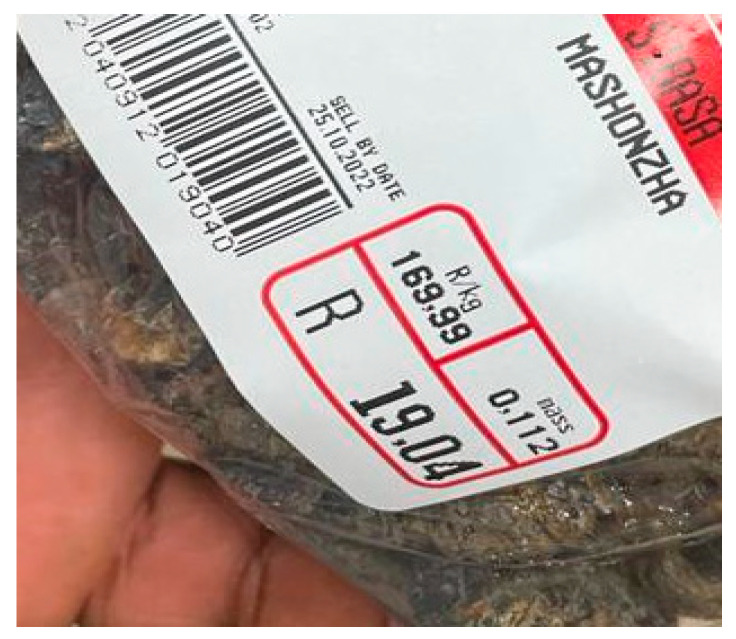
*Cirina forda* sold at one of the biggest grocery stores in South Africa (Nemadodzi, L.E., January 2023).

**Figure 4 foods-12-02184-f004:**
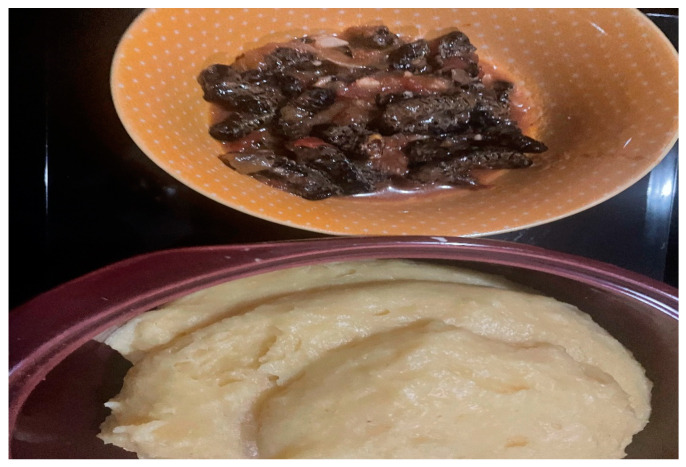
A common South African meal with edible caterpillars (Nemadodzi L.E., October 2022).

**Table 1 foods-12-02184-t001:** Costs of mopane worms as reported in 2014. The exchange rate has since changed quite dramatically, as of 2022.

Countries		Amount		References
**Botswana**	**Currency in Pula**		**United State currency in Dollar**	
32 kg bag	10		1.15	[51]
26,000–30,000 kg	200,000		23,019.99	[51]
**South Africa**	**currency in Rands**		**US$**	
1 kg	9		0.89	[52]
1 kg	25–41		2.50–4.00	[53]
80 kg bag	1200–1600		118.28–157.70	[54]
5 L bucket	150		14.78	[54]
10 L bucket	300		29.57	[54]
**Zimbabwe**	**Currency in Dollar**		**United State currency in Dollar**	
1 kg	158		0.42	[52]
1 kg	210		0.56	[52]
1 kg	123		0.33	[52]
1 kg	357		0.96	[52]
1 kg	230		0.62	[52]
1 kg	500		1.34	[52]
1 kg	300		0.80	[55]

## Data Availability

Data used in the study is included in the manuscript, and authors are cited both in the text and in the reference lists.

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
