# Peer review of "The Use of Gonimbrasia belina (Westwood, 1849) and Cirina forda (Westwood, 1849) Caterpillars (Lepidoptera: Sarturniidae) as Food Sources and Income Generators in Africa"

_foods, 2023, doi:10.3390/foods12112184_

Round 1

Reviewer 1 Report

This manuscript is very important first step in comprehensive studying of edible caterpillars and their importance for indigenous people and entomophagy in Africa. In future work authors may aim on another aspects.

The Title does not seem to be very accurate, only lines 205-235 and in the Recommendation lines 483-486 are devoted to trees. I suggest either changing the name to something more general – such as “overview of mopane” (because the text contains a lot about development and nutritional values ​​and sales) or omit from the text everything that does not relate to trees or, add additional information about trees - toxicity of leaves, map of the predominant occurrence of individual tree species.

Line 4 - Lufuno Ethel Nemadeodzi – prorably a typo, should be Nemadodzi?

Use of abbreviations - usually the full name of the plant or animal is given at the first mention in the text and in brackets the common name and the abbreviation that is used later in the text, e.g. Gonimbrasia belina (mopane worm, G. belina) Burkea africana tree (B. africana ). E.g.: line 22 - Go. belina should be G. belina (please correct in the whole manuscript).

Line 42 - Hymennoptera should be Hymenoptera

Line 77 - Colosphospermum mopane should be Colophospermum

Figure 1 – the quality of the photos is unfortunately poor; the individual stages cannot be distinguished on them. Since the individual stages of development are not dealt with further in the text, I suggest completely omitting them, or replacing them with photographs, even taken by someone else, but of high quality.

Line 291 - Figure 3: Cirina Forda – should be forda.

Figure 2,3 – Please credit the author of the photos.

Chapter 3.2 I suggest converting into a table, add a map of the area (Africa? Sub-Saharan Africa?) with the approximate occurrence of individual tree species. It would be clearer.

Table 1-6 I suggest modifying the title – e.g. instead of “Table 1 provides the cost…”  use just “Table 1: The cost …” Tables should be referenced in the text above the table.

Table 2-6 –It would be very useful to add Recommended Daily Dosage.

Author Response

Dear Reviewer, 

Thank you very much for your positive and detailed critiques. Below are the points addressing the comments: 

Title

The reviewer is thanked for the comment as the authors acknowledge that the title was not reflecting the content of the manuscript. The title has changed as per your advice as now reads as follows "The use of Gonambrasia belina (Westwood,1849) and Cirina forda (Westwood, 1849) caterpillars (Lepidoptera: Sarturniidae) as food source and income generator in Africa".

Line 4: typo on the last name edited, now reads Nemadodzi.

Use of appropriate abbreviation on G. belina corrected throughout the entire document.

Line 42: Hymenoptera corrected.

Line 77: Colophospermum corrected.

 The reviewer is thanked for the comment on Figure 1. The photos in Figure 1 and all other Figures were revisited and replaced where the quality could be improved. It was decided that some of the photos in Figure 1 was not of good quality and Figure 1 was therefore changed to only include the highest revolution photos and to include the most relevant photos as per the suggestion of the reviewer.  

Line 291: changed to Cirina forda

Figure 2, 3: Photo credit was added to all the Figures in the manuscript.

3.2 Thank you very much for your suggestion.   A map of Africa/ Sub-Saharan with occurrences of host trees of G. belina and C. forda is currently not available as these are widely spread across the continent. We opted for the available scientific literature with citations which can easily be found by other researchers and scientists. However, the suggestion has identified a gap which should be explored and investigated in future studies.

Tables 1-6 were all reviewed, and the content carefully considered. Some of the tables were combined which now provides easy comparison between the data available for the different species.

Table 2-6 The daily recommended dosage was included in lines 270-273. 

Reviewer 2 Report

Evaluation of the article: Identification of threes that serve as a host for Gonimbrasia belina and Cirina forda caterpillars, a delicacy in Africa: A Review.

The article, in general, is well structured, however the content does not coincide with the title, since for example the topics are widely explained: food security and nutritional value and these are not mentioned in the title, but they are indicated in the keywords. Furthermore, it is not explained why the title includes the word delicacy.

TITLE

Regarding the title, I suggest indicating in the scientific names the author's last name and the year in which said species were described and indicating Lepidoptera: Saturniidae.

Abstract

Line 13 correctly separate the word diet

Line 22 I suggest writing in all cases only G. belina.

KEYWORDS

Line 26 The keywords do not match the title of the article eg endophytes,  developing countries , income generation.

INTRODUCTION

Line 42 correctly spell the word Hymenoptera.

I suggest integrating the paragraph of lines 61 and 62 There are……..are edible in line 53 where the order Lepidoptera is mentioned and in line 53 eliminating which include edible saturniids.

Line 77 In this line and in all scientific names I suggest writing the name of the author and the year in which the species is described, for example Colophospernum mopane, line 98 Burkea africana, Cirina forda etc etc.

Line 88 separate March &.

MATERIALS AND METHODS.

Correct the corresponding spacing lines 120 to 132.

Line 140 correctly separate the word arid.

Line 163 I did not find in the references to Raletsana 2022 or it is a personal communication??  to clarify, the same situation occurs in line 166 with Rasiwelame.

Line 189 repeats Cirina forda is univolttine, this aspect is indicated in line 188.

Line 203 clarify in the title of figure 1, the author is mentioned Nemadodzi 2002 and this reference does not appear in the list, or Nemadodzi only did the collection????

Line 248 include table 1 in the text, write the title, and discuss the price changes separately. In addition, as the magazine is international, write down the meaning of BWP ZAR and ZWD

Line 257 write Fig. 2a and 2b.

Line 260 correctly separate the word different.

Line 268 Figure 2a and Figure 2b line 275 caption is the same edit.

Line 279 I did not find in the references to Masethe V 2022  or is it a personal communication???

Line 285 correctly separates the word figure.

Line 311 Correctly separate the word issues, the same in line 615 caterpillar.

Line 333 clarify the paragraph "edible insects have been described as one of the solutions to curb food security?? and reduce poverty by FAO”.

Line 349 Figure 4 is not related to the information presented on line 349, therefore I suggest including the following additional words AND THEY EVEN PREPARE IT AS DELICACY after the word Woodland on line 348.

Integrate Table 2 in the text, correct the title of the table and discuss the results presented in another paragraph.

Tables 2, 3, 4, 5 are not mentioned in the text, therefore it is necessary to integrate them into the paragraphs that comprise lines 365 to 373 and 381 to 393.

Line 406 Table 2 does not have a title, and I suggest integrating the loss of nutrients such as sodium, potassium and iron in the discussion of the text.

Line 406-407 In Table 2 I suggest pointing to one of the rows processed caterpillars and removing converted .

Line 412, In Table 3 I suggest writing down the title, and indicating what results were obtained on a dry basis and discussing the results in another paragraph.

In Line 414 the source 84 is indicated, whose title also corresponds to fatty acids that are not mentioned in YOUR table.

Line 416 Enter the title of Table 4, discuss the results in another section and indicate in which units the minerals are reported.

Line 418 reference 84 does not mention minerals???

Line 420 in Table 5, write the title, and write down if the reported results correspond to a dry base or a wet base ,( you  are mixing the results of the wet base and the dry base, which is incorrect), in your table if the sample has 83.11% of water, the dry matter that is calculated by difference corresponds to 16.9 %, therefore I recommend modifying this table and calculating the corresponding results on a dry  basis  or  wet basis and/or only reporting the results on a dry basis.

Line 426 Integrate Table 6 in the text, write the title and discuss the results.

Also, if the fonts and drives are the same then I suggest merging table 4 and table 6.

Line 434 the title of reference 87 is not related to its text, therefore it is necessary to explain why they are used in disease management.

Line 461-462, in your text polyunsaturated fatty acids are not mentioned, therefore, I suggest including them.

Line 471-475, how do you plan to guarantee mass production and avoid over-exploitation?

REFERENCES.

Line 519, 525, 530, 666, etc. Is it correct to point   in some references et al.???

Line 535 writes the abbreviated name of the journal and in other cases, for example 546,547, 591, the name is complete, therefore I suggest homogenizing all references.

Line 548 pages are missing.

Line 559 review and/or correct the reference, does it have volume 0? Biotropic ATBC?????.

Line 562 The abbreviations of the names of the journals generally do not have points.

Line 571 write Sarturniidae and J. res…

Line 571 write Westwood

Line 575 write the scientific name in italics

Line 583 is missing the article title.

Line 602 pages are missing

Line 602 missing pages

Line 626 delete pp.

Line 639 correct the reference for example the title does not match and it does not have volume or pages.

Line 647 is missing co-author Vantomme P.

Line 681 is missing the name of the journal and the volume

Line 682 is missing the name of the journal and the volume

Line 686 writes van and Line 688 Van, I suggest to homogenize.

Line 688 The official name of the reference  Annu.  Rev. Entomol. You  write it with points and in lines 690 the points are missing. I suggest homogenizing in all cases of the references line 697, 699, 700, 701 etc.

Line 692 write A.O.

Line 694 , correct the missing reference title of the journal, i volume,  number etc.

Author Response

Dear Reviewer,

Thank you very much for your positive and detailed critique.  Below is the point- by- point feedback on the amendments as per your suggestions.

Title

The reviewer is thanked for the comment, as the authors acknowledge that the title was not reflecting the content of the manuscript. The title has been changed as per your advice and as now reads   "The use of Gonimbrasia belina (Westwood, 1849) and Cirina forda  (Westwood, 1849) caterpillars (Lepidoptera: Sarturniidae)  as food source and income generator in Africa.

Abstract

Line 13: diet correctly separated.

Line 22: Go. belina  and in all cases replaced with G. belina

Keywords

endophytes deleted, however income generation left as is since these caterpillars are not only meant for consumption but play a huge role as a source of income generation. Developing countries replaced with Africa. 

Introduction

Line 42: Hymenoptera correctly spelled.

Line 61 and 62 merged with line 53, saturniids eliminated as per your suggestion.

Line 77 and Line 98: Author and the year in which the species was described added.

Line 88: March& separated.

Materials and Methods

Line 120-132 spacing line corrected to correspondence.

Line 140: arid correctly separated.

Line 163 personal communication added on Raletsana and Rasiwelame.

Line 189 Repeated ''Cirina forda is univolttine deleted.

Line 203: Thank you for your question,  Nemadodzi  2021-2022 is the first author of the current manuscript who collected Cirina forda and captured the life cycle.

Line 248 BWP, ZAR, ZWD written in full in the table.

Line 257: Fig 2a and 2b added.

Line 260 fixed.

Line 268 Figure a and Figure 2b caption edited.

Line 285 the word figure automatically separated as changes were made.

Line 279: personal communication on V. Masethe added.

Line 311: the word issues corrected, line 615 the word caterpillars fixed.

Line 333 sentence clarified.

Line 394 the words "and they even prepare it as a delicacy" added after woodland as per your suggestion. 

Table 2 integrated in-text and a paragraph discussing the results also added.

Table 2, 3, 4, 5 added intext previously line 365, 373, 381, 393.

Line 406, title on table 2 added, sodium and potassium added on the discussion. 

Line 406-407 the word converted removed.

Line 412: Title on Table 3 written and discussion below the table included. 

Line 414 Total polyunsaturated fatty acids, total monounsaturated fatty acids and total saturated acids added on the table.

Line 416: The content and titles of the tables were reviewed and corrected.  Also, results discussed on separate section (see line 442-450).

Line 418 edited to mineral composition.

Line 461-462: polyunsaturated fatty acids are mentioned in Line 454, Line 455, Line 467 and line 470.

Line 471-475 the answer to your question has been added.

References

Line 519, 525, 530, 566 all authors added.

Line 546, 547, 591 references homogenized by replacing the journal abbreviation. 

Line 583: article title added,

Line 602: pages added.

Line 626: pp deleted. 

Line 639: reference corrected.

Line 647: co-author Vantomme P added. 

Line 681, 682 name of the journal and volume added.

Line 686, 688: Van replaced with van, and homogenized.

Line 688, 690, 697, 699, 700, 701 homogenized as per your suggestion

Line 692 A.O edited.

Line 694, thank you for your advice, the old reference has been replaced with the correct reference with the journal title, volume and number. 

Reviewer 3 Report

General comments: The study aimed to identify host treesfor Gonimbrasia belina and Cirina forda caterpillars. However, only a small section of the manuscript focused on this objective, with the rest delving into economic value and nutrient composition. The literature search appeared poorly planned, and the findings were not new or comprehensive. The manuscript lacked coherence and was poorly organized. All figures had a low quality while several tables were directly copied from the references. It is also unclear whether the authors have obtained permissions to use the figures and tables. Due to the abovementioned reasons, I cannot recommend the publication of this manuscript.

Specific comments

Line 2: The title is not reflective of the content.

Lines 23-25: These topics digress from the focus of the paper.

Line 42: This is an overestimation.

Line 65: Spell out the genus name when it first appears in the text and then abbreviate as G. belina. Make changes throughout the manuscript.

Line 115: Need to justify why this topic is important.

Lines 120-122: Some of these are not typical databases used for literature search.

Lines 122-125: Using these strings is not a good searching strategy for this topic.

Lines 344: Again, this overestimated number needs to be updated.

Line 406: The authors should summarize and perform higher level synthesis of information instead of copying and pasting the whole table from the reference.

Line 407: Which caterpillar?

Author Response

Dear Reviewer, 

Thank you very much for you detailed and positive critiques. Below are the point-by-point response (s) to your concerns and comments. 

Answer: The manuscript was thoroughly reviewed, and the authors agreed that the title could have been more descriptive of the content. The title was subsequently changed to " The use of Gonimbrasia belina  (Westwood, 1849) and Cirina forda  (Westwood, 1849) caterpillars (Lepidoptera: Sarturniidae) as a food source and income generator in Africa". The title is now more descriptive of the content of the manuscript. 

The Figures were revised and replaced where the quality could be improved. All the photos were taken by the main author of the manuscript and has been added. It was decided that some of the photos in Figure 1 was not of good enough quality and Figure was therefore changed to only include the highest resolution photos.  All ethical procedures were followed as stated in the ethics declaration at the end of the manuscript when photos were taken.   

The tables were also reviewed and integrated to add more value to the manuscript. For instance, by combining the information for the two caterpillars in one table, it is easier to compare the data available for each of the caterpillar species. 

Figure 3: Is a photo of edible caterpillars bought at a grocery store as supporting evidence in comparison of the unit price. 

Figure 4: is a South African meal prepared and cooked by Nemadodzi L.E who is the first author of this manuscript. 

Although the photos were captured by a high-quality camera, we apologise for the low quality that might have caused discomfort. 

Line 2: The title has been rephrased as indicated on general comments).

Line 22-25: deleted as per your advice.

Line 42: The information provided here, is from a reputable source, and as it is clearly referenced, we state that the information is not our own, but provided by the respective authors. As there are no other information available, we cannot determine if it is an overestimation or not, but it is important to provide information that is currently available in literature and therefore, it was included in the manuscript.   

Line 65: changes incorporated on Gbelina throughout the entire document. 

Line 115: This manuscript aims to provide a balanced view on the importance, but also the topical issues related to the use of insects as food. As much as edible caterpillars forms an important source of food in Africa, there is still reluctance to consumption. For instance, in South Africa, certain tribes and races do not consume these caterpillars due to ''cultural myth'' that they are worms. Most of the tribes would prefer chicken/beef/fish to caterpillars as a source of food. Moreover, edible caterpillars are perceived as food for the poor, hence adaptability and more awareness on the nutritional benefit offered by these caterpillars is of vital importance. The manuscript therefore covers not only the importance of edible caterpillars as food, but also the impact of cultural and indigenous believes and the role it plays in food security over time.

Line 120-122 and Lines 122-125:  University of South Africa is the Institution of higher learning at which the three authors are affiliated with, with one of the most comprehensive libraries globally.  As stated in the manuscript, firstly the UNISA library was used to screen the suitable information, and thereafter, the other "unconventional " databases and search engines were also used and resources consulted. However, the authors did find that different search engines provide different resources and therefore, the use of the different and not commonly used databases and search engines were found to be useful. By using different databases and using different keywords, and additionally string searches, including related keywords, the authors believe that we have retrieved all the relevant literature related to the topic. All literature cited in this manuscript was acknowledged in the form of intext citation and in the reference list and included based on the inclusion criteria as stated in the manuscript. 

Line 344: Again, the information was reported by [19], which is a published book and therefore it was deemed a reputable source. The authors acknowledge that as research evolves, future studies will have updated information to add on the already reported. The authors believe that it is important to include this information, as there are no other sources available to compare data with. 

Line 406: The reviewer is thanked for this comment. The information provided in the tables were reviewed and is not presented to allow for easy comparison of the different species.

Line 407: G. belina and C. forda added.

Round 2

Reviewer 2 Report

The authors corrected the article according to the observatioons previously made.

Author Response

Dear Reviewer, 

Thank you very much for your detailed and positive critique. Below is the point-by-point feedback on the amendments as per your suggestion.

The reviewer is thanked for the comment, as the author acknowledge that the title is not reflecting the content of the manuscript. The title has been changed as per your advice and now reads as follows '' The use of Gonimbrasia belina (Westwood, 1849) and Cirina forda (Westwood, 1849) caterpillars (Lepidoptera: Sarturniidae) as food source and income generator in Africa''.

Reviewer 3 Report

The authors have addressed some of my comments. Regarding the estimation that more than 2 billion people consume insects, the author who made that statement himself has acknowledged that the number was an overestimation (Van Huis et al., 2022). This must be updated.

Reference: Van Huis, A., Halloran, A., Van Itterbeeck, J., Klunder, H., & Vantomme, P. (2022). How many people on our planet eat insects: 2 billion?. Journal of Insects as Food and Feed, 8(1), 1-4.

Author Response

Dear Reviewer, 

The authors thanked the reviewer for the positive comment and feedback. 

The recent update has been included and reference cited intext also included in the reference list (see lines 41-43) and updated reference list now reads reference number 5.